# Attention to Women’s Sexual and Reproductive Health at the Street Outreach Office

**DOI:** 10.3390/ijerph191710885

**Published:** 2022-09-01

**Authors:** Nayara Gonçalves Barbosa, Thaís Massita Hasimoto, Thamíris Martins Michelon, Lise Maria Carvalho Mendes, Gustavo Gonçalves dos Santos, Juliana Cristina dos Santos Monteiro, Flávia Azevedo Gomes-Sponholz

**Affiliations:** 1Department of Maternal-Child Nursing and Public Health, Faculty of Nursing, Federal University of Juiz de Fora, Juiz de Fora 36036-900, Brazil; 2Nursing School of Ribeirão Preto, University of São Paulo, Ribeirão Preto 14040-902, Brazil; 3Public Health Nursing Post-Graduate Program, Nursing School of Ribeirão Preto, University of São Paulo, Ribeirão Preto 14040-902, Brazil; 4Department of Maternal-Child Nursing and Public Health, Nursing School of Ribeirão Preto, University of São Paulo, Ribeirão Preto 14040-902, Brazil

**Keywords:** women’s health, sexual and reproductive health, homeless women, gender inequity, health inequity

## Abstract

The aim of study was to understand care strategies for homeless women, focusing on aspects of sexual and reproductive health from the perspective of Street Outreach Office strategy professionals. This is a qualitative study carried out in a large city in the countryside of São Paulo, Brazil. Data were collected from December 2020 to April 2021 through semi-structured interviews, and the results were obtained through content analysis and thematic modality. Nine health professionals participated in this study, and the analysis of the interviews allowed identifying three thematic categories: (i) being female and sex on the streets (violence and oppression); (ii) gynecology as a gateway to comprehensive care for homeless women; and (iii) pregnancy, puerperium and motherhood in the context of the streets. This study contributes to the reflection of professional practices within the Street Outreach Office, allowing the understanding of challenges in assisting homeless women, aiming to raise awareness of professionals and services that make up the health care network.

## 1. Introduction

Homeless women belong to one of the most vulnerable population groups, in a context permeated by violence [1,2], marginalization, exploitation, stigma, invisibility [3], prejudice, gender and social rights inequalities [4]. Homelessness constitutes a public health concern in the Brazilian context. In 2012, homeless people in Brazil numbered 96,560; by 2020 the number had grown by 230%, reaching 221,869 [5]. In the same tendency, today the epidemiology has shifted to include higher proportions of women, who represent 18% of homeless people and are concentrated in the 18–45 years of age group, in the reproductive period [6].

Life on the streets exposes women to challenging situations that involve the integrity of their bodies, sexuality and their health [2,6]. Due to their adverse life situation, homeless women are exposed to violence [1,2], at high risk of sexually transmitted infections (STI) and unplanned pregnancies [7,8,9], contributing to the perpetuation of the cycle of poverty [10], misery and social exclusion [2]. Sex on the streets is linked to forms of survival and can be currency for protection, guarantee of a place to sleep or for the acquisition of money. However, this condition hurts human dignity and shows the violation of human, sexual and reproductive rights. The specificity and complexity of homelessness are major challenges for various sectors and services of society, including the health system [11]. The Brazilian National Policy for the Homeless Population (PNHP) represents a milestone in the defense of the right to health of the Brazilian homeless people [12]. The Street Outreach Office team’s work stands out here as a health care strategy for the homeless population, whose objective is to articulate and provide comprehensive health in the territory [13], with the incorporation of harm reduction, disease prevention and promotion and health education practices [11,14].

The Street Outreach Office articulates the access of its users to primary health care services that are gateways to the health system in Brazil. In addition, they operate on an itinerant basis, developing actions locally, in street scenarios [14] by building a therapeutic setting based on bonding and welcoming the homeless population [14,15].

Considering gender as a category of social analysis [16] that permeates the health–disease-care process, it is assumed that Street Outreach Office professionals recognize that the situation of women on the street permeates all dimensions of social relationships, and whose roots lie in gender inequalities and direct strategic actions to promote comprehensive health care for this population are provided. In this sense, the gender perspective described by Scott [16] was considered as a theoretical framework for this study, since it can help to understand the process of construction of subjective identity, culminating in a process of differentiation and distinction between masculinity and femininity. The women and the gender issues among the homeless population are generally under-researched, especially in low-income countries [2]. The comprehensiveness of health care practices for homeless women is essential to improve the assistance and value the gender perspective in care. These challenges are aligned with the United Nations Sustainable Development Goals, including: End poverty in all its forms everywhere; Ensuring healthy lives and promoting well-being; Achieve gender equality and empower all women and girls; Reduce inequality; and Promote just, peaceful and inclusive societies.

Thus, this study aimed to understand the care strategies for focusing homeless women on aspects of sexual and reproductive health from the Street Outreach Office professionals’ perspective.

## 2. Materials and Methods

### 2.1. Ethical Aspects

The study was approved by the Research Ethics Committee linked to the Brazilian National Research Ethics Committee, in accordance with the principles of Resolution 466/2012 of the Brazilian National Health Council (Conselho Nacional de Saúde). Additionally, the study was authorized by the Municipal Health Department of the studied city. Participants signed the informed consent form (ICF) and participated voluntarily in the study.

### 2.2. Study Design and Place

This is a qualitative study, developed in the Street Outreach Office modality II, composed of a multidisciplinary team who works on an itinerant basis in several locations in a large city in the countryside of the state of São Paulo, Brazil. The Consolidated Criteria for Reporting Qualitative Research (COREQ) was used to guide the study [17].

### 2.3. Study Participants

All professionals who were active in the Street Outreach Office during the data collection period were invited to participate in the study. All agreed to participate in the study, with the participation of nine professionals, by signing the ICF. The professionals interviewed were psychologists, nurses, nursing technicians and community health workers. Professionals who made up the permanent team and those who worked in the Street Outreach Office due to the COVID-19 pandemic were included, since all of them were involved and engaged in the delivery of health care services even in such a troubled period.

### 2.4. Data Collection

Data were obtained from December 2020 to April 2021 through semi-structured interviews, with an average duration of 45 min, carried out by a nursing student after previous training. The interviews were scheduled in advance with the participants and took place in a private place in a health service center. Safety measures were taken to prevent COVID-19 transmission, including conducting the interview in an airy place, wearing masks, at a minimum distance of one meter [18] and with hand hygiene.

Each interview took place in a flexible way, respecting the desire and availability of each participant to talk about the subject and share their experiences. The interviews were audio-recorded in MP3 format and fully transcribed. To guarantee the secrecy and confidentiality of the professionals who participated in the research, they were identified by the letter “P”, followed by an ordinal number, according to the order in which the interviews were carried out (P1–P9).

The interview script contained two parts: the first referring to participant sociodemographic characteristics. In the second part, there were the following guiding questions: What are the health needs presented by the homeless women you care for? What is the Street Outreach Office team’s proposal in approaching women on the streets?

### 2.5. Data Analysis

The interview contents were arranged from the full transcription of all recordings, preserving their originality. After transcribing the interviews, a thorough reading was carried out, followed by successive material reading and detailed exploration, aiming at identifying the relevant aspects to answer the research objectives.

The analysis of discourses was based on thematic content analysis [19], which consists of searching for meanings that represent a frequency of units of meaning to reach the object of study. We sought to find categories that were significant expressions according to which the speeches’ content was organized [19]. Data pre-analysis, exploration, inference and interpretation were thoroughly performed [19].

Sexual and reproductive health issues were recurrent in Street Outreach Office professionals’ responses and were heavily explored in the data analysis. Data were analyzed considering the gender perspective as mentioned above [16].

## 3. Results

Nine professionals participated: two psychologists, three nurses, three community health workers and one nursing technician. Most were female (6/9—67%). Their average age was 52 years (SD = 7.33, range = 33–63 years) with job tenure ranging from 1 month to 11 years. Most (7/9–78%) reported training, such as harm reduction courses, STI/AIDS and training carried out by the Street Outreach Office team’s in their daily work in health.

Three themes were identified in the data analysis: being female and sex on the streets (violence and oppression); gynecology as a gateway to comprehensive care for homeless women; and pregnancy, puerperium and motherhood in the context of the streets.

### 3.1. Being Female and Sex on the Streets: Violence and Oppression

The participants stated that in the context of life on the streets, they faced women exposed to a diversity of types of violence. The reports of daily violence experienced by homeless women who are seen at the office were present in the professionals’ narratives (Table 1).

Hearing reports that the street is a place of struggle for survival is a constant for these professionals, who, in their day-to-day work, serve women who can trade sex for food, money, drugs, and “protection”. Eventually homeless women avoid permanence in certain places with prostitutes as they can suffer violence from being beaten by these prostitutes, revealing different situations of inequality even among women. Additionally, the homeless women are susceptible to sexual assault, and the abuse of alcohol and psychoactive substances contributes to enhancing their vulnerability.

Time on the streets is a physical and mental traumatic event for women and has a significant impact on their behavior. Mainly, women change their attitudes that lead to the reproduction of violence and aggression as a strategy for self-defense and survival on the streets. This behavior can be a barrier to care because they do not allow the approximation of the health workers. Thus, the care of homeless women is a challenge. Building bonds and gaining the confidence of these women is an essential part of the care process.

### 3.2. Gynecology as a Gateway to Comprehensive Care for Homeless Women

Participants stated that homeless women face barriers in accessing health services (Table 2). In this sense, the professionals from Street Outreach Office provide this access, through gynecology care, which most women seek. Gynecological care is not only the opportunity for entry into the health system but also a health necessity. The gynecological consult includes cervical cancer screening, STI screening using rapid tests, attention to sexual violence and pregnancy diagnosis. It also denotes the design of strategical actions aimed at this population, such as the availability of long-acting reversible contraceptives (LARC) such as subdermal implants.

Harm reduction includes providing increased access to condoms and lubricants for sex as well as other inputs and instructional materials. Despite the supply distribution, such as condoms and lubricants, professionals recognize that there are no guarantees about the use of condoms on all occasions and about safe sexual practices for women living on the streets.

### 3.3. Pregnancy, Puerperium and Motherhood in the Context of the Streets

Pregnancy on the streets is a great challenge for Street Outreach Office teams, especially due to the difficulty in carrying out and monitoring prenatal care (Table 3). In the professionals’ work, welcoming pregnant women, often users of psychoactive substances, is carried out in addition to guidance, testing, vertical infection transmission prevention, transport and guarantee of a place in the maternity ward.

The identification of pregnancy can occur late, amid difficulties in linking women to prenatal care and monitoring the dyad.

Attention to infections in the gestational period is part of Street Outreach Office team’s repertoire of actions, with the development of actions aimed at timely diagnosis, treatment, and prevention of vertical transmission. In this scenario, challenges are identified in follow-up of women to complete the instituted therapeutic regimen. Care in the puerperal period also represents a great challenge for Street Outreach Office teams in homeless women’s care.

The welcoming posture, the importance of dialogue, the participation of women in decision-making and bond formation between professionals and women stand out.

Motherhood on the streets presents numerous difficulties which can result in the separation of the dyad due to the situation of vulnerability in which the woman/mother finds herself, which can interfere with care conditions and in the maintenance of children’s well-being. In these cases, the Guardianship Council is activated after birth and the child can be referred to family care or foster care institutions.

## 4. Discussion

The present study identified that Brazilian’s Street Outreach Office workers recognize the conditions of the vulnerability of homeless women marked by gender, which represents a transgression of women’s human, sexual and reproductive rights [2].

The gender perspective here is helpful to decode meanings, to understand narratives about human interactions, and to facilitate understanding of the role of organizations and institutions in maintaining and reproducing interpersonal power relationships in the daily lives of the women studied [16]. Therefore, understanding the meaning and the repercussions of these experiences for homeless women is essential to build an individual care plan with them. The gender perspective, social class, race and generation should be considered during care [20].

Workers approach the street territory (geographic, cultural and existential) and seek to understand its dynamics and complexity [21]. The activities realized on the streets and public places allows the development of care in an unconventional space marked by environmental exposition (sun, rain, hot, cold, odor and dirt) [21]. Additionally, the Street Outreach Office is an access gateway to the Brazilian public health system and intersectoral networks to this marginalized and invisible population in the search for guaranteeing their rights [15].

Regarding the homeless women, they are physically and sexually assaulted, leading to unwanted and high-risk pregnancies [15]. An American study, realized in San Francisco, carried out with 300 homeless women, found that the main forms of violence experienced in the prior 6 months were psychological violence (87%), physical violence without weapons (48%), physical violence with the use of weapons (18%) and sexual violence (18%) [1].

The majority of homeless people who were victims of physical or sexual violence did not receive any type of assistance, demonstrating the weakness in access to health care for this population [1,14]. It is noted that the lack or delay in assisting the victims of sexual violence has a negative impact on their physical health due to the risk of acquiring infections, the occurrence of pregnancies, plus emotional and social repercussions, including stress, depression, suicide and substance abuse [14]. In this way, it becomes imperative to provide comprehensive care to homeless women regarding their vulnerability to violence.

In the Street Outreach Office, gynecological care represents an opportunity for comprehensive attention to homeless women’s care, allowing the reception and care of health needs and the provision of care in an expanded perspective.

The Street Outreach Office develops inter and transdisciplinary care, with intra and intersectoral care networks impacting the lives and health of the homeless [22]. Primary health care (PHC) is a powerful and strategic instance utilized to provide the care of homeless people. Related to women’s health in PHC are developed actions related to screening of cervical and breast cancer, care of gynecological problems, STI screening and treatment, reproductive planning, antenatal care for low-risk pregnancy, puerperal care and attention to sexual and domestic violence [22].

PHC develops educational, preventive and mental and physical health promotion activities for women of all ages, including attention to climacterium [23]. However, there are difficulties in following homeless women and providing continuity of care. Mainly, women move to other places or cities, promoting the rupture of care.

In our study, despite the offer of supplies for harm-reduction and condoms by Street Outreach Office teams, it is difficult to monitor the effective use of prevention and protection methods by homeless women, especially due to the complexity of their experiences in the context of the streets. The vulnerability of homeless women in relation to lower use of condoms highlights the historically unfavorable gender inequality as well as exposure to sexual violence, which hinders their autonomy to make decisions, including the negotiation of protected sex [7].

Implementing health education strategies supports women in making informed and safe decisions about their sexuality and reproduction and provides subsidies to encourage such decisions. A potential solution to reduce the rates of unintended pregnancies in homeless women refers to access to counseling regarding contraceptive methods and receiving information in understandable language in addition to greater access to effective methods of contraception, including LARC, if desired by the women [24]. In Brazil, there is distribution of contraceptive methods such as masculine and feminine condoms, hormonal pills and injectable (combined and progesterone), intrauterine devices and permanent methods [22]. The offer of subdermal implants to women in vulnerable situations is not a consensus in Brazil, although some cities, such as Ribeirão Preto, have adopted this strategy.

Despite the development of prevention and counseling strategies, the pregnancy of homeless women is a reality in Brazil. One study showed that no woman wanted to become pregnant while homeless. On the other hand, there are homeless women who wish to experience motherhood or even have ambivalent feelings about being a mother even when living on the street [25]. Precarious living conditions on the streets disfigure an adequate space for pregnancy and the exercise of motherhood [26]. In addition to this, barriers to accessing services represent risk factors for the dyad’s health [27]. Additionally, homeless women who become pregnant have higher rates of complications in the pregnancy–puerperal cycle [10] as well as the risk for maternal mortality, which is asserted in the conditions of social inequality of these women.

Among the challenges mentioned by the Street Outreach Office teams, the difficulties of following up women for treatment of infections stand out, impairing maternal and fetal prognosis. Due to the difficulty of linking homeless women to prenatal care and puerperal follow-up, the approach in the Street Outreach Office requires strategies that allow effective assistance to pregnant women’ specificities living on the streets [28], in addition to health promotion and disease prevention actions during pregnancy and continued care in the puerperal period [29]. In this direction, the training of health professionals for welcoming homeless women is salutary, as barriers in prenatal care, discontinuity and challenges of care in the puerperal period were also reported by the Street Outreach Office teams [29].

It is worth noting that, on the streets, the complexity of motherhood also involves the fear of losing custody of their children in addition to the shame for being on the streets with their children or for having already lost family power over their children [25]. This reality was observed in the experience of professionals in homeless women/mother care. 

It was detected in Street Outreach Office team narratives that the formation of bonds occur during homeless women’s care. The therapeutic relationship between professionals, users and service practices contribute to expanding access [15] and attending women’s health from an integral perspective. In this direction, using soft skill technologies in the Street Outreach Office’s daily work stands out, with the use of relational tools in health care that provide the establishment of trust and dialogue between users and professionals, in addition to favoring the construction of bonds, commitment, and accountability [1]. Such elements are fundamental for the comprehensive care of women who are on the streets, guaranteeing their right to health and human dignity.

Conducting this research only with Street Outreach Office teams, without analyzing the homeless women’s perspective, can be considered a limitation of this study. However, the method used allowed the understanding of the challenges in assisting this population, aiming to raise awareness of the professionals and services that make up the health care network.

## 5. Conclusions

The study contributed to the reflection of professional practices within the Street Outreach Office. Exposure of women to violence, the need for gynecological and obstetric care as an opportunity to link women to comprehensive care, difficulties related to the pregnancy–puerperal cycle and the exercise of motherhood in the context of the streets were evidenced.

The implementation of a different perspective, considering the principle of equity for the health of women living on the streets, is a sine qua non condition for guaranteeing their human, sexual and reproductive rights.

## Figures and Tables

**Table 1 ijerph-19-10885-t001:** Workers’ narratives related to violence against homeless women.

Themes	Sub-Themes	Narratives
Violence and oppression	Vulnerability to violence	“There is this issue of marginalized women, women who suffer violence and women living on the streets are totally vulnerable.” P1 “(...) many of them stay on the street, sleeping in squares, get a companion and a dog to stay safe, not to be abused and even then it ends up happening. (...)” P6
	Situations of violence	“Sexual violence from partners or others on the street, because she sells her body in exchange for drugs. She suffers violence from security professionals. We have reports that there are some professionals who hit women especially. They suffer violence from other women, especially sex workers, who think that the person on the street will fight for their piece, their field. So, there is a lot of violence on the streets for women.” P4 “But they have their partners, they all do. There is one that was arrested these days, because she killed her partner. Then she was arrested... yeah, they have sexual partners with others.” P2 “…when she is using her drug on the street, she sells her body in order to get this drug, this alcohol, her self-esteem, her self-worth is greatly diminished. She is aware of it, she is aware of it. And many times, for her not to be attacked, she already attacks people who come close.” P4

**Table 2 ijerph-19-10885-t002:** Some workers’ narratives related to homeless women’s care developed in a Brazilian Street Outreach Office.

Themes	Sub-Themes	Narratives
Gynecology as a gateway to comprehensive care for homeless women	Activities developed by Street Office Outreach	“The needs, the biggest, that we find a lot, are in relation to sexually transmitted diseases, we address this a lot. We also offer contraceptive methods. The worker gives a general approach, like a medical clinic, even hygiene and health issues, we bring them, sometimes we take a bath here, they offer a hygiene kit, the necessary guidelines.” P8 “...we have, as a root, the distribution of condom inputs, cocoa butter, female condom... self-test for HIV. So, we use it for the distribution of supplies, so we can bond with this woman and be able to make the necessary appointments, usually a gynecologist, a CAPS (service at the Psychosocial Care Center), usually the female homeless person, usually the woman, is more specific related to the gynecologist, and from the gynecologist we can cover other specialties. Then comes dentistry, other specialties, but the gynecologist is like a gateway for women in health.” P3 “Our proposal is to bring health, knowledge, right, about sexually transmitted diseases, family planning, right, self-care.” P2 “...sex workers, female homeless people have priority, for example, for the placement of Implanon, which is a contraceptive.” P3
	Building a bond	“As you gain their trust, you bring them here, they take a shower, we give them clothes, register, go to the social worker, and if there is any gynecological complaint, we schedule something. If there is an open schedule, they are already seen on the same day by the gynecologist here, trying to convince them of (use) Implanon, (performing) the pap smear, so here there is a very good service.” P2
	Limitations of practices	“But they take condoms, do they use them? Well, I do not know. I can’t tell you, but we provide, right? But I don’t know if they use it, they say they use it.” P2

**Table 3 ijerph-19-10885-t003:** Some narratives from participants related to pregnancy, puerperium and motherhood in a homelessness situation.

Themes	Sub-Themes	Narratives
Pregnancy, puerperium and motherhood in the context of the streets	Lack of antenatal care	“... pregnant women without prenatal care, not knowing how their health is, suddenly exposed to some type of STI and not knowing or not caring, so it is important to follow up even this dyad, right?” P1 “All pregnant women, when discovered, it has to be done, I call it mothering, we have to take them to collect serologies, we have to take them for prenatal care, we have to take them for tests, we have to take them to do ultrasound. We follow her until she arrives at the maternity ward, always offering her some devices that we have, then in the ninth month we have to guarantee a spot (...) we guarantee it at the Clinical Hospital, she comes back to be able to have the baby. So, with pregnant women, we have that, but not all of them accept to go to a care facility” P4 “Congenital syphilis is a very serious problem, because you have to treat it... and treatment, as they are not frequent, it’s hard to find, you can’t give them three doses of benzathine penicillin G. You do the first one, the second one we find them 4 km from the city, and sometimes they are fugitives from the police, they are in debt, and then you can’t give their name anywhere, so you lose documents, those kinds of things. All this we have to go through with them, we go through it all because of the need they have.” P6
	Activities developed with homeless pregnant women	“As for pregnant women, we talk about everything, talk about prenatal care, talk about exams, if they have a disease, if they have a street disease, if they have anything like that, we take them and take care of them.” P6 “You have to know how to talk, you have to know how to respect their time, because if we arrive with authority, you won’t get anything. You get things with them by respecting and talking, explaining what the reasons are, the reasons that many times the babies that are going to be born, they are not to blame. So, we are there to help her as a pregnant woman and the baby that is coming to come a strong, beautiful, healthy child.” P5 “When we are able to help those who refer them to the Basic Health Unit, they have to order all the tests on the first day they arrive, because when they come back, the baby may have already been born, you know? Where was prenatal care done? There is no record.” P6
	Puerperium and motherhood	“If she is pregnant, we do not receive care before delivery, and even after the puerperium, it is a very serious thing... so, I have seen pregnant women, mothers who went to the street bleeding, badly, cesarean scar bleeding, purulent, without this care...” P4 “They are on crack a lot, there are babies that we follow up, they are born, they go to families, some families take them, others don’t, they go to the Guardianship Council, they go for adoption.” P5 “Usually, she is separated from her baby. These children, which is not our decision, it is a decision of the guardianship council... they need psychological support, they need care.” P4 “I was remembering Alice (fictional name). She was a homeless pregnant woman. We followed her in antenatal care, she gave birth and she came back to the street. The baby was carried to a foster care institute. So, 20 days later we received the news: Alice was dead from a delivery complication. No one wanted to identify Alice’s body in the institute of legal medicine. We must do it, go there and identify that she was Alice (...).” P4

## Data Availability

The data presented in this study are available on request from the corresponding author. The data are not publicly available due to data privacy following the approval by the Research Ethics Committee of the University of São Paulo at Ribeirão Preto College of Nursing, Brazil.

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
