# Peer review of "Attention to Women’s Sexual and Reproductive Health at the Street Outreach Office"

_ijerph, 2022, doi:10.3390/ijerph191710885_

Round 1
Reviewer 1 Report
First of all, thank you for allowing me to review this manuscript. I think this topic is relevant. Also, the approach may be the correct one to properly understand the insights about sexual care in the homeless women population. However, I´ve got a few suggestions to improve the manuscript and finally be suitable for publishing:
- I would recommend an English check, noted a few grammatical errors, nothing major, but especially when it comes to qualitative research the words need to mean what they are supposed to. Thus I believe an English check will improve this.
- I´ve noted mistakes when referencing, some of them appear just as a number without brackets probably not linked with referencing gestor. I May advise reviewing it and also maybe include a few more references that empower the findings, especially in the discussion.
- The weaknesses/limitations should not be included in the conclusion, as well as its justification. The conclusion, must be the final summary of the findings, which may have clinical practice implications/applications, but nothing else.
- When approaching the limitations (in the discussion part of the manuscript) I strongly advise treating more in-depth the reason why homeless women didn't take part in the interviews, at least they should be considered somehow.
- In addition, the job tenure range (1 month, which essentially is 1/12 year to 11 years) may condition the experiences of the healthcare professionals, although this may be argued in terms of training and courses undertaken.
- Keywords should include "homeless women" instead of people on the street.
I think if those changes are addressed the manuscript is worth publishing.
Author Response
Dear reviewer 1
Please see the attachment.
Best regards,
The authors.

Reviewer 2 Report
The author's present a study that has the potential to give innovative insight into the practices of Sao Paolo's Street Outreach Office. The overall presentation of the manuscript lacks an organization of the literature/ scholarly context and the study's findings that would allow the reader to identify:
*specificities of the research context versus (mainly cited) US-contexts and the respective differences in practices (if there are; if there are not, this needs to be mentioned/contextualized in some way)
*the gains of knowledge that this study presents for the specific context of Sao Paolo. We mainly find references to studies in other geographical contexts, where infrastructures etc. are different. Thus, I wonder to what extent the comparison is applicable to the context of the study?
*how the gathered data can be reflected in a way that draws out/summarizes new insights what professional practitioners experience and what the take-aways are from this study
*who the audience of the study is - does it provide something else but a collection of findings? Are there specific recommendations that could be made to help practitioners better address issues? Are there specific offices or decision-makers involved in how the situation is at the moment? If so, can the author's engage the contextual factors that would help readers understand why it is the way it is?
Firthermore, interview data is organized in a way that is very hard to read and understand. Not only are there differences in font style but it appears the explanation of a phenomenon is given before the contextual evidence is presented. PErhaps choosing a table format is more suitable. In addition, "results" are too general in the respective section. The author's should engage with the interview data directly instead of making very broad and unreferenced statements. Perhaps, the author's can cut down the studies on the US, which make up a significant portion of the paper but are unrelated to the geographical context, and there is no connection made between the two. I suggest preferencing contextual knowledge over that from another geographic region, or drawing better connections between the two and state why the relation is needed.
The author's should spell-check the text. In addition, sentences are often filled with two-four ideas. I suggest avoiding run-on sentences for clarity. Cumbersome sentence constructions in the introduction and throughout the entire discussion part make it hard to read and follow.
There is also no clear definition of the perspective of gender taken in the paper. Are you talking about cis-women only, or are you taking into account transgender women as well? If so, are there any aspects that the data reflects concerning the findings, and whether the differential notions of gender affect what has been found in the interviews? A closer link to gender studies and health literature that mentions concepts with which the authors work explicitly are needed to follow the interpretation that the authors provide in the discussion.
Throughout the text, crucial aspects are introduced that should be clear from the set-up of the paper: e.g. SUS principles, an explanation of the acronyms CCM and LARC. Instead of using the term "speeches", authors could say "statements" when refering to the contributions of study participants.
It further appears that some information that belongs in the introduction appears in the discussion part, e.g. the introduction of the Street Outreach Office. Some information on the materials collected is repeated in the results section. That should be in the materials section and condensed.
Author Response
Dear reviewer 2,
Please see the attachment
Best regards,
The authors.

Round 2
Reviewer 1 Report
The English is improved, did not find any mistakes or grammatical error at this point now, however don't have this option to tick in the reviewing platform. The references are fixed. Limitations are included and they address all of the study's weaknesses. Keywords match study content. From all of the recommendations made and after changes, I honestly think the investigation is worth publishing.
Author Response
We greatly appreciate your considerations.
Best regards,
The authros